# Extracts from Environmental Strains of *Pseudomonas* spp. Effectively Control Fungal Plant Diseases

**DOI:** 10.3390/plants11030436

**Published:** 2022-02-05

**Authors:** Valentina Librizzi, Antonino Malacrinò, Maria Giulia Li Destri Nicosia, Nataly Barger, Tal Luzzatto-Knaan, Sonia Pangallo, Giovanni E. Agosteo, Leonardo Schena

**Affiliations:** 1Department Agraria, University of Reggio Calabria, 89122 Reggio Calabria, Italy; librizzi.valentina@libero.it (V.L.); giulia.lidestri@unirc.it (M.G.L.D.N.); sonia.pangallo@unirc.it (S.P.); geagosteo@unirc.it (G.E.A.); lschena@unirc.it (L.S.); 2Department of Marine Biology, Leon H. Charney School of Marine Sciences, University of Haifa, Haifa 3498838, Israel; nbarger@univ.haifa.ac.il (N.B.); tluzzatto@univ.haifa.ac.il (T.L.-K.)

**Keywords:** *Penicillium*, *Botrytis*, *Colletotrichum*, *Alternaria*, *Monilinia*, post-harvest diseases

## Abstract

The use of synthetic chemical products in agriculture is causing severe damage to the environment and human health, but agrochemicals are still widely used to protect our crops. To counteract this trend, we have been looking for alternative strategies to control plant diseases without causing harm to the environment or damage to our health. However, these alternatives are still far from completely replacing chemical products. Microorganisms have been widely known as a biological tool to control plant diseases, but their use is still limited due to the high variability in their efficacy, together with issues in product registration. However, the metabolites produced by these microorganisms can represent a novel tool for the environment-friendly management of plant diseases, while reducing the issues mentioned above. In this study, we explore the soil microbial diversity in natural systems to look for microorganisms with the potential to be used in pre- and post-harvest protection against fungal plant pathogens. Using a simple workflow, we isolated 22 bacterial strains that were tested both in vitro and in vivo for their ability to counteract the growth of common plant pathogens. The three best isolates, identified as members of the bacterial genus *Pseudomonas*, were used to produce a series of alcoholic extracts, which were then tested for their action against plant pathogens in simulated real-world applications. Results show that extracts from these isolates have an exceptional biocontrol activity and can be successfully used to control plant pathogens in operational setups. Thus, this study shows that the environmental microbiome is an important source of microorganisms producing metabolites that might provide an alternative strategy to synthetic chemical products.

## 1. Introduction

Evidence that a lot of the chemical products used in agriculture are harmful to the environment and human health has been around for several decades [1,2,3]. This has generated a huge response from consumers, farmers, scientists, and policy makers, which has promoted the use of agricultural products obtained with low or no chemical inputs; at the same time, this has fostered research on alternative strategies to prevent damage from pests and pathogens [4]. Microorganisms have been found to be potential competitors of chemical pesticides. However, microbial-based products still struggle to penetrate the market and replace synthetic molecules, mainly because of the high variability of their action, availability, and persistence, together with issues in product registration [5,6,7,8].

Microorganisms have a long story of being used to control pre-harvest and post-harvest pathogens [9,10,11]. These microbes belong to several taxonomical groups [12] and exploit a wide variety of mechanisms to contrast the development of fungal plant pathogens, including the production of metabolites, enzymes, and siderophores; the competition for nutrients and space; and the modulation of plant physiology (e.g., the induction of systemic resistance) [13]. Several potential biocontrol agents have been studied, and a few have been commercialized [14], but their marketability is still insufficient because of their low efficiency and reliability compared to chemical products [11]. *Bacillus*, *Pseudomonas*, and *Paenibacillus* are all bacterial genera that include strains with biocontrol activity [7,15,16], and they are commonly found to be associated with plants and soil. Although microbial biocontrol agents are still not competitive on the market, recent studies on plant and soil microbiomes show that the diversity of these microbial communities is incredibly high [17,18]. Thus, there is still a wide potential pool of microorganisms in natural environments that might offer new opportunities for the biocontrol of pre- and post-harvest diseases.

Screenings for biocontrol organisms have been mainly performed on plant tissues or the rhizosphere of diseased plants, resulting in isolates with high capacity for contrasting plant pathogens [7]. Biocontrol organisms can be isolated from the environment and screened in mass to select those with antagonistic action against the target pathogens. In this framework, the extreme diversity of the environmental microbiome [17] might represent a powerful source of microorganisms with the potential to be used in pre- and post-harvest protection. Within these complex microbial communities, the competition between species and strains is very high, and some microbes can diversify to reduce the fitness of competing species (e.g., antibiotics, fungicides), so they increase their competitive ability and occupy new niches. These microbial strains and the metabolites they produce thus have a high potential to be isolated and used to counteract the growth of pathogens that damage agricultural products. Here, we test the hypothesis that the environmental microbiome is a source of microorganisms that can be isolated, selected, cultivated, and used as a source of active metabolites to control fungal plant diseases in real-world applications.

## 2. Materials and Methods

### 2.1. Study Overview

In this study, we exploited the soil microbial diversity in natural environments in order to discover novel bacterial isolates as a potential source of bioactive compounds that can be used in plant protection (Figure 1). We started by collecting soil from different environments with low anthropic impact, creating a microbial wash from each of these soils and plating each microbial wash together with a suspension of spores of *Penicillium digitatum*. This allowed us to select 22 bacterial isolates that showed antifungal activity. All these isolates were then tested for their antifugal activity both in vitro (dual-culture assay) and in vivo (co-inoculating fruits together with a pathogen) against 5 different fungal pathogens (Table 1). Subsequently, we selected the 3 isolates that showed the best results in both assays, and we taxonomically identified them as species of the bacterial genus *Pseudomonas*. To better investigate the molecules with antifungal action, we produced an alcoholic extract from each bacterial isolate, characterized by using metabolomics and tested for: (i) efficacy to prevent diseases caused by fungal pathogens, (ii) induction of systemic resistance, (iii) curative activity of early-stage infections, (iv) efficacy in field trials. Data analysis was performed using R 4.1.0 [19] with the packages *lme4* [20] and *car* [21]. The modeling strategy is detailed for each trial described below. The package *emmeans* [22] was used to extract post hoc contrasts.

### 2.2. Isolation of Potential Biocontrol Agents

Potential biocontrol agents were isolated from soil collected in five different forests located in southern Italy (Appendix A). At each location, we collected soil from the top 5 cm below the litter, bulking together 10 subsamples of ∼100 g from 10 randomly selected spots within a radius of 50 m. The soil was then sieved through a 2 mm mesh to eliminate large debris (e.g., leaves, roots, stones) and stored at 4 °C until further processing.

We selected potential biocontrol agents by growing together a microbial mix obtained from soil and a high-density suspension of spores of *P. digitatum* (10${}^{6}$ spores/mL). The microbial mix was obtained from each location (Appendix A) by mixing 10g of soil with 100 mL of water using a magnetic stirrer for ∼20 min. Each soil was then serially diluted to a 1:1000 ratio with sterile water. This process was repeated three times for each location, and the three subsamples were bulked together before plate inoculation. The spore suspension of *P. digitatum* was obtained by harvesting conidia from isolates grown on PDA (Potato Dextrose Agar) plates at 20 °C for 7–10 days. Conidia were collected with a spatula, suspended in sterile distilled water, filtered through a double layer of sterile gauze, and vortexed for 1 min to ensure uniform mixing. Conidia from this stock solution were counted using a Thoma cell and then diluted to a density of 10${}^{6}$ spores/mL. A total of 100 µL of each soil microbial mix was then co-inoculated with 100 µL of *P. digitatum* spore suspension on PDA, evenly distributed with a sterile spatula and incubated at 25 °C for 7 days. The plates were periodically observed to identify bacteria producing inhibition halos, yielding 125 bacterial colonies that were then isolated on Nutrient Broth Agar (NBA). These isolates were grown on NBA plates for 7 days at 24 °C, and then roughly screened to select isolates with different morphological features (e.g., different colony shape, color, and size). This yielded 22 bacterial isolates that were then used for the experiments below.

### 2.3. In Vitro Antifungal Activity

The 22 bacterial isolates were first tested to estimate their capacity to inhibit in vitro the growth of three common plant pathogens: *Botrytis cinerea*, *Alternaria alternata*, and *Phytophtora palmivora*. Tests were run using a dual-culture assay [7], growing together combinations of each bacterial isolate with one of the three pathogens, and then estimating the growth reduction of the fungal pathogen. Each PDA Petri dish was inoculated with ∼0.16 cm^2^ agar plug with fungal mycelium (obtained from a pure culture of the pathogen) placed on one side of the plate. On the opposite side of the plate, a single bacterial isolate was inoculated in a single spot by transferring cells from the pure cultures using a sterile needle. Each pathogen-bacterial isolate combination was replicated 3 times, and a set of 3 plates for each pathogen without the bacterial isolate served as control. Plates were incubated at 22 °C for 5 days in the dark, and the radial growth of the pathogen was then measured for each plate. The inhibition of fungal growth was then estimated as percentage reduction compared to control plates. Data were fit to a linear model, including *isolate ID*, *pathogen ID*, and their interaction as fixed factors.

### 2.4. Preventive Antifungal Activity of Live Bacteria

The same 22 bacterial isolates were tested for their ability to contrast fungal infection on fruits. For this assay, we used six different fruit–pathogen combinations (Table 1). Fruit surface was sterilized by immersion in a 2% sodium hypochlorite solution for 2 min, washed with tap water, air-dried, and fixed onto polypropylene panels using a double-sided tape [23]. Fruits were kept 1–2 cm apart to avoid nesting, and wounded to a uniform and standard depth of 3 mm using a nail with diameter of 1 mm. Wounds were inoculated by applying 10 µL of bacterial suspension or water (negative control). The bacterial inoculum was prepared by growing each isolate in Erlenmeyer flasks with 5 mL of Luria–Bertani (LB) broth on a rotary shaker at 100 rpm, for 48 h at 25 °C. The cells were harvested by centrifugation (1100× *g* for 15 min), rinsed twice with water, and resuspended in 5 mL of distilled water. All bacterial suspensions were diluted at a 1:10 ratio before inoculation. After two hours from bacterial inoculation, the same wounds were inoculated with 10 µL of fungal conidial suspension obtained from each fungal species, as reported above for *P. digitatum*, and diluted with distilled water in order to have 5 × 10${}^{4}$ conidia/mL (*P. digitatum*) or 10${}^{5}$ conidia/mL (*P. expansum*, *A. alternata*, *Colletotrichum acutatum*, and *B. cinerea*). These concentrations were chosen according to preliminary trials. A pomegranate peel extract (PGE) at a concentration of 6 mg/L was used as a positive control [23,24]. Tests were performed on 15 fruits (apple, tomato) or 30 fruits (grape, olive, tangerine) for each treatment.

After the inoculations, fruits were maintained at room temperature (22 ± 2 °C) in plastic boxes containing wet paper to ensure high relative humidity, and the presence or absence of fungal growth was recorded after 7 days. Within each host–pathogen combination, disease incidence for groups treated with a bacterial isolate was compared to the negative control by fitting a generalized linear model, using *isolate ID* as the fixed factor and specifying a binomial error distribution. According to the results from these assays (see below), all the following trials focused only on isolates B01, B05, and B09.

### 2.5. Preventive Antifungal Activity of Extracts

Alcoholic extracts from isolates B01, B05, and B09 were prepared to test the efficacy of metabolites produced by these bacteria in protecting harvested fruits from fungal diseases. Each bacterial isolate was grown in an Erlenmeyer Flask containing 20 mL of Nutrient Broth (NB) for 2 days at 28 °C on a rotary shaker. Then, bacterial suspensions were transferred into 500 mL glass bottles containing 80 mL of absolute ethanol. Bottles were vigorously shaken for 5 min and then kept overnight on a rotary shaker. The extract was aliquoted in 1.5 mL tubes and transferred to a rotary evaporator overnight. Once dry, the pellet (approximately 0.02 g) was transferred to a single tube and suspended in 66.7 mL of absolute ethanol and then stored at −20 °C until use.

We tested the efficacy of these extracts to prevent fungal disease in fruits artificially inoculated with pathogens using three host–pathogen combinations (Table 1): apricot—*M. fructicola*, tangerine—*P. digitatum*, and tomato—*B. cinerea*. The fruits selected had a uniform size, and they were surface-sterilized and wounded, as described above. Each wound received 10 µL of bacterial extract diluted with sterile distilled water, with final concentrations at 1.5, 3, 6, 12, 24, and 36 mg/L. The fruits treated with a 10 µL water solution containing 12 mL/L of ethanol (similar to the ethanol concentration inoculated in fruits treated with 36 mg/L of extract) were used as a negative control. Once dry (∼2 h), the wounds were inoculated with a 10 µL spore suspension containing 5 × 10${}^{4}$ (*P. digitatum*) or 5 × 10${}^{5}$ (*B. cinerea* and *M. fructicola*) conidia/mL. Tests were run on 45 (apricots), 54 (tangerine), or 180 (tomato) fruits per group. The fruits were incubated at room temperature (22 ± 2 °C), in plastic boxes containing wet paper to ensure high relative humidity, and after 7 days, they were scored for the presence or absence of fungal growth. Data were then fit to a generalized linear model using the *isolate ID* and *dose* (and their interaction) as fixed factors and specifying a binomial error distribution.

We used an untargeted metabolomics approach to characterize the composition of the alcoholic extract of the three isolates B01, B05, and B09. Analyses were conducted according to Luzzatto-Knaan et al. [25]. Briefly, liquid chromatography was carried out on a UPLC ultimate 3000 dionex system with a C-18 column (Phenomenex 1.7 µm C18 50 × 2.1 mm) in the following conditions: A- ACN: 0.1% FA; B- H20: 0.1% FA. Flow: 0.5 mL/min; 0 min: 90% B, 0.5 min: 90% B, 3 min: 50% B, 8 min: 1% B, 11 min: 1% B, 11.5 min: 90% B, 12.5 min: 90% B. Mass spectrometry measurements were carried out on a Bruker Maxis impact QTQF system in an ESI positive mode. The method used was tune positive MS/MS, with the following MS conditions: Ionization mode: ESI positive; Capillary: 4000 V; Corona: 4000 nA; Nebulizer: 2 Bar; Dry Gas: 5 L/min; Dry temp: 200; Vaporizer Temp: 450; Active: 5; Exclude after: 4; Release after: 0.5 min; Absolute Threshold: 213 cts; Relative Threshold: 0%; Spectra rate: 1 Hz; Precursor Ion list: exclude. For MS2, Auto MS/MS transitions were used. LC-MS raw data files were converted to the mzXML format by Compass DataAnalysis 4.2 software (Bruker Daltonics). Feature detection was performed by GNPS molecular networking based on MS/MS spectra.

### 2.6. Induction of Resistance

We tested the efficacy of extracts from the B01, B05, and B09 bacterial isolates in inducing resistance in fruits artificially inoculated with pathogens. For this test, we used three host–pathogen combinations (Table 1): apricot—*M. fructicola*, grape—*B. cinerea*, and tomato—*B. cinerea*. The fruits selected had a uniform size, and they were surface sterilized, as described above. In this case, each fruit was wounded twice at a distance of ∼2 cm. We first inoculated the bacterial extract into one wound at a concentration of 1.5, 12, and 36 mg/L. Then, after 24 h, the fungal pathogen was inoculated into the other wound by applying 10 µL of a suspension containing 5 × 10${}^{5}$ conidia/mL. In this way, we were able to test whether the bacterial extracts could induce resistance mechanisms in fruits, reducing their susceptibility to fungal pathogens. A group of fruits inoculated with a water solution containing 12 mL/L of ethanol served as a negative control. Fruits (45 per group) were incubated at room temperature (22 ± 2 °C), in plastic boxes containing wet paper to ensure high relative humidity, and after 7 days, they were scored for the presence/absence of fungal growth. Data were then fit to a generalized linear model, using *isolate ID* and *dose* (and their interaction) as fixed factors and specifying a binomial error distribution.

### 2.7. Curative Effects

We tested the curative effect (control of pre-existing infections) of extracts from the B01, B05, and B09 bacterial isolates. Tangerine fruits were selected to have a uniform size, surface-sterilized as described above, wounded, and inoculated with 10 µL of a suspension containing 5 × 10${}^{4}$ conidia/mL of *P. digitatum*. After 24 h from inoculating the pathogen, we inoculated 10 µL of each extract at the concentration of 1.5, 12, and 36 mg/L. This time frame was considered enough for the germination of the conidia of *P. digitatum* and the starting of the infection process. A group of fruits inoculated with a water solution containing 12 mL/L of ethanol served as a negative control. Fruits (45 per group) were incubated at room temperature (22 ± 2 °C), in plastic boxes containing wet paper to ensure high relative humidity, and after 7 days, they were scored for the presence/absence of fungal growth. Data were then fit to a generalized linear model, using the *isolate ID* and *dose* (and their interaction) as fixed factors and specifying a binomial error distribution.

### 2.8. Control of Post-Harvest Rots on Olives

After selecting and assessing the biocontrol potential of our bacterial extracts, we tested their efficacy in situations close to real-world applications. In this first trial, we tested whether extracts from the isolates B01, B05, and B09 were able to control post-harvest fruit decay caused by *Colletotrichum* spp. on olive fruits (Table 1). The fruits were collected in a commercial orchard located within the Gioia Tauro plain (southern Italy) where *Colletotrichum* species (mainly *C. acutatum s. str.* and *C. godetiae*) are endemic, selected to be uniform in size and ripeness, avoiding fruits with lesions and any symptoms of fungal disease. Fruits were then divided into 4 groups (∼1500 fruits each) and treated with one of each of the three isolates or water as a negative control. The olives were left to dry for ∼2 h at room temperature, transferred to plastic boxes, and incubated at room temperature (22 ± 2 °C). The boxes contained wet paper to ensure high relative humidity. After 7 days, we scored each fruit for the presence/absence of fungal rots, and we scored the decay severity using an empirical scale, according to the amount of fruit surface showing symptoms: 0 (no rot), 1 (<25%), 2 (25–50%), 3 (50–75%), 4 (>75%). This allowed us to calculate McKinney’s index [26], which considers the disease incidence (presence/absence) together with its severity. Data were then fit to a linear model, using *treatment* as a fixed factor.

### 2.9. Control of Post-Harvest Rots on Sweet Cherries

In this second trial, we tested the efficacy of the alcoholic extracts of the B09 bacterial isolate to control post-harvest diseases (mainly *M. fructicola*) on sweet cherries. We selected only this bacterial isolate because it is the one consistently showing the best results across all the previous trials. For this experiment, we collected two cherry varieties (Ferrovia and Giorgia) in a commercial orchard located in Puglia (southern Italy). Fruits with lesions or disease symptoms were discarded. The fruits (50 per group) were dipped into a solution containing the alcoholic extract from isolate B09 at different concentrations (6, 12, and 24 mg/L) or tap water (negative control), dried at room temperature for 2 h on blotting paper, placed in plastic trays, covered with plastic sheet, and stored at 2 ± 1 °C. Incidence and severity of decay were evaluated after 14 days of cold storage, using the same empirical scale described above. Data were then fit to a linear model, using *treatment* as fixed factor.

### 2.10. Control of Powdery Mildew in a Commercial Grapevine Orchard

In this third trial, we tested the efficacy of the alcoholic extract from isolate B09 in controlling powdery mildew (*Uncinula necator*) in grapevine commercial orchards. Field trials were conducted in Cirò (southern Italy), in a 10-year-old vineyard cultivated with the variety Gaglioppo (Table 1) and managed using an organic farming approach. The trial consisted of three blocks, each divided into 4 groups, which were treated using: (i) the alcoholic extract from isolate B09 at a concentration of 24 mg/L; (ii) a pomegranate peel extract (PGE) at a concentration of 6 mg/L as a positive control [24,27]; (iii) chemical products as normally used for the rest of the vineyard as second positive control (see below); (iv) no treatment (negative control). Treatments were applied following the farm schedule. This trial was performed in 2021, when the severity of powdery mildew was rather low, so the farm scheduled only two treatments on 6 June (copper-based Coprantol Hi Bio 2.0, Syngenta) and on 5 July (sulfur-based Tjovit Jet, Syngenta). All the treatments above were applied on the same days. The incidence and severity of the powdery mildew were scored on 3 July and 29 July on leaves and bunches. The leaves were divided into mature and young in order to differentiate those already present at the time of the treatment from those that developed later (and did not directly receive the treatment). We evaluated symptoms on 72 bunches and 90 young and mature leaves for each treatment, scoring their severity on an empirical scale, according to the amount of fruit/leaf surface showing symptoms: 0 (no damage), 1 (<20%), 2 (20–40%), 3 (40–60%), 4 (>60%). In addition, we tested whether our treatments would influence the sugar content of grapes. The data were gathered on 31 August after collecting 48 berries per replicate (144 for each treatment) and measuring the total soluble solids (°Brix) with a refractometer. Data were then fit to a linear model, using *treatment* as a fixed factor.

## 3. Results

### 3.1. In Vitro and In Vivo Antifungal Activity

First, we tested the efficacy of the 22 bacterial isolates in reducing the growth of three plant pathogens (*B. cinerea*, *A. alternata*, and *P. palmivora*) in vitro and their efficacy on reducing post-harvest fruit disease in six different host–pathogen combinations (Table 1).

The in vitro assays suggest that each isolate has a different efficacy in reducing the growth of fungal pathogens, and this depends on the identity of the pathogen itself (isolate × pathogen interaction F = 98.63; df = 42, 132; *p* < 0.001; Appendix A). While no bacterial isolate was able to efficiently inhibit all three pathogens, isolates B02, B03, B09, and B17 were able to reduce the growth of both *B. cinerea* and *P. palmivora* by >40%, while isolates B13 and B21 were able to reduce the growth of both *B. cinerea* and *A. alternata* by >40% (Appendix A).

In the in vivo assays, regardless the host–pathogen combination (Table 1), all the fruits in the negative control group (ethanol) developed the fungal disease, while no fruit in the positive control group (pomegranate peel extract, PGE) showed symptoms of decay. Thus, we decided to test the efficacy of each bacterial isolate against the negative control, within each host–pathogen combination (Appendix A). Given that all the fruits from the negative control group developed decay, differences would represent the efficacy of the isolate in controlling the fungal pathogen. The isolates numbered from B12 to B22 were not able to significantly reduce the incidence of decay in any host–pathogen combination (Appendix A); therefore, we decided to focus on the isolates numbered from B01 to B11. All these isolates were able to significantly reduce decay in olives inoculated with *Colletotrichum acutatum s. str.* (Appendix A). Most of the isolates were able to control *B. cinerea* in tomato (except B01, B04, B10, and B11) and grape (except B06, B07, and B11), as well as *P. digitatum* in tangerine (except B07, B08, and B11). On the other hand, just a few isolates were able to effectively reduce decay by *P. expansum* in apple (B05, B09), and symptoms by *A. alternata* in tomato (B01, B02, B05, B09). In general, isolates B05 and B09 were able to significantly reduce decay in all host–pathogen combinations, and were used for the following tests together with isolate B01. As reported above and in the Appendix A, isolates B01, B05, and B09 were identified as *Pseudomonas* spp.

### 3.2. Preventive Antifungal Activity of Extracts

Given the efficacy shown by isolates B01, B05, and B09 in protecting fruits from fungal pathogens, we then tested the hypothesis that their action was mainly driven by the metabolites they produce. Thus, we prepared an alcoholic extract from each isolate and tested their efficacy in preventing the development of fungal decay in tangerine (*P. digitatum*), tomato (*B. cinerea*), and apricot (*M. fructicola*).

In tangerines, we observed the effect of the treatment (B01, B05, B09, or control; $\chi^{2}$ = 207.16; df = 3; *p* < 0.001), but no effect was driven by dose or the treatment x dose interaction (*p* > 0.05). Indeed, while all fruits in the control group showed symptoms of decay, no fruits treated with the B01 or B09 extract showed any sign of decay, and only 3.71% of fruits treated with the B05 extract showed symptoms. Similarly, we found an effect driven by the treatment ($\chi^{2}$ = 207.16; df = 3; *p* < 0.001) when testing our extracts on apricots against *M. fructicola*, but no effect was driven by dose or the treatment x dose interaction (*p* > 0.05). Moreover, in this case, all control fruits showed signs of decay, while only 4.44% of fruits treated with isolate B01 showed the disease by *M. fructicola*; no fruits treated with the B05 and B09 extracts showed any symptoms. When testing the efficacy of the alcoholic extracts on tomatoes against *B. cinerea*, we found an effect driven by the treatment ($\chi^{2}$ = 406.95; df = 3; *p* < 0.001) but no effect driven by dose or the treatment x dose interaction (*p* > 0.05). Additionally, in this case, all untreated fruits developed fungal decay, while all three isolates were able to reduce the incidence of the fungal disease to 15.5% (B09), 17.22% (B05), and 25% (B01), with no differences between extracts, as suggested by the pairwise contrasts (*p* > 0.05).

### 3.3. Induction of Resistance

We also tested the extracts from the B01, B05, and B09 isolates for their ability to induce resistance against post-harvest pathogens by inoculating extracts and pathogens into two different wounds on each fruit. Results show that all three isolates were able to fully protect the apricots against *M. fructicola* at any dose, while all control fruits developed fungal decay ($\chi^{2}$ = 202.44; df = 3; *p* < 0.001). Similarly, the B01 and B09 extracts fully protected the tomatoes against *B. cinerea* regardless the dose, while 22.2% of the fruits treated with the B05 extract showed symptoms, and all the control fruits developed decay.

### 3.4. Curative Effects

In addition, we tested the curative effects of our extracts by inoculating tangerine fruits with spores of *P. digitatum* and treating them with the extracts after 24 h, allowing time for the spores to germinate and begin to infect the fruit. Furthermore, in this case, the results suggest an effect driven by treatment ($\chi^{2}$ = 159.48; df = 4; *p* < 0.001), with all negative control fruits showing symptoms, all positive control fruits (PGE) not showing any symptoms, only 17.7% of fruits treated with B05 and B09 extracts showing decay, and 26.6% of those treated with the B01 extract showing signs of infection.

### 3.5. Metabolomics

Given the efficacy of the extracts from all three isolates (B01, B05, and B09), we used an untargeted metabolomics approach to try to identify the molecules that might be responsible of their antifungal activity. By removing compounds identified in the blank samples and in the bacterial growth medium, we found 337 molecules associated with our bacterial extracts. The majority of these molecules did not find a match in the database, while we found phenylpropanoids and polyketides (3.56%), benzenoids (2.97%), organoheterocyclic compounds (1.78%), organic nitrogen compounds (0.89%), lipids and lipid-like molecules (0.3%), and organophosphorous compounds (0.3%). All the extracts shared 71 compounds, of which 11 were putatively annotated as: epitestosterone-like, tolcapone-like, avobenzone-like, pheniramine N-Oxide-like, neoeriocitrin-like, carbamazepine-like, N-succinylmexiletine-like, altenusin-like, phenazine-1-carboxylic acid-like, sphingosine-like, and velutin-like. Extracts from isolate B01 were characterized by a unique signature of 18 compounds, of which none was accurately identified. Similarly, 22 metabolites were uniquely identified in extracts from isolate B05, but we were not able to identify any of them. Extracts from B09 show a unique blend of 36 metabolites, of which we were able to identify only one as cryptotanshinone-like.

### 3.6. Control of Post-Harvest Rots on Olives

After testing the extracts from the B01, B05, and B09 isolates in controlled conditions, we ran three trials to simulate their use in simulated operative conditions. The first trial assessed whether the three extracts could help in controlling post-harvest rots on olives caused by *Colletotrichum* spp. The control group resulted in 100% of fruits showing decay, while those treated with PGE resulted in no fruits showing any symptoms. Analyses suggest an effect driven by the treatment (F = 200.25, df = 4, 10, *p* < 0.001), and post hoc contrasts suggest differences between the three extracts and both the control and PGE groups (*p* < 0.001), but with no differences between the extracts (*p* > 0.05).

### 3.7. Control of Post-Harvest Rots on Sweet Cherries

Given the high performance of extracts from isolate B09, we focused this and the next trial only on this isolate. Moreover, in this case, we compared its performance to a control group and a treatment with PGE. Results (Figure 2) show an effect driven by the treatment for both the varieties Ferrovia ($\chi^{2}$ = 402.89; df = 2; *p* < 0.001) and Giorgia ($\chi^{2}$ = 232.3; df = 2; *p* < 0.001). In both cases, post hoc contrasts show similar results: the control group had the highest incidence and severity of diseases, followed by the B09 extract, and PGE.

### 3.8. Control of Powdery Mildew in a Commercial Grapevine Orchard

As the last trial, we tested the efficacy of the extract from isolate B09 in controlling powdery mildew in field conditions, comparing it with PGE, a chemical treatment, and an untreated control. In this case, we tested the effects on disease incidence and severity in young leaves, old leaves, and fruits at two time points. The results (Figure 3) suggest no effect of any treatment on young leaves (F = 3.07; df = 3, 8; *p* = 0.09), while all treatments reduced the disease symptoms in old leaves compared to control (F = 7.94; df = 3, 8; *p* = 0.008), but no differences were recorded between treatments (*p* > 0.05). During the first sampling on fruits, we did not observe any effect driven by the treatment (F = 1.84; df = 3, 8; *p* = 0.21), while on the second sampling, we observed a treatment effect (F = 8.13; df = 3, 8; *p* = 0.008) mainly driven by the reduction in disease severity in the PGE group compared to the control (*p* = 0.006). Finally, we also tested whether the treatments impact the amount of sugars in fruits at harvest, and we did not find any significant effect (F = 1.14; df = 3, 8; *p* = 0.38).

## 4. Discussion

In this study, we isolated microorganisms from the environment and selected a group with a high potential for the biological control of pre- and post-harvest pathogens. In addition, we found that alcoholic extracts from bacterial isolates have good performance against fungal pathogens, also when tested under conditions simulating real-world applications.

As a first step, we used a simple method of selecting microorganisms from the environment, which yielded 125 bacterial isolates. These were further reduced to 22 isolates, which were then tested for their in vitro and in vivo activity to contrast the growth of fungal plant pathogens. The in vitro tests showed that all the 22 isolates were able to significantly reduce the growth of at least one of the three fungal pathogens we used (*A. alternata*, *B. cinerea*, *P. palmivora*), and the outcome strongly varied according to the isolate–pathogen combination. On the other hand, the in vivo tests showed that roughly half of our isolates were able to significantly reduce the incidence of decay, although only two isolates were able to consistently prevent fungal growth in all six trials. These isolates (B05 and B09) were further selected for more focused bioassays together with an additional isolate (B01). All three isolates were identified to belong to the genus *Pseudomonas*, although we were not able to identify them at the species level. This bacterial genus hosts a wide diversity of species with different ecological and functional roles [28,29], and they can be pathogens or beneficial associates to plants [29,30]. Strains of *Pseudomonas* have been previously studied for their potential in controlling plant pathogens [31]. Given the high diversity of *Pseudomonas* strains in soils [32] and their known biocontrol activity [33], it is not surprising that the strains with the highest performance belong to this genus. Indeed, previous studies reported strains of *Pseudomonas* with in vitro antifungal activity against *A. alternata* [34], *B. cinerea* [35], and *P. palmivora* [36]. In addition, members of the genus *Pseudomonas* have been reported to reduce the incidence of fungal disease in agricultural products during the post-harvest phase. For example, strains of *Pseudomonas* isolated from disease-suppressive composts successfully protected blueberry fruits against *A. alternata* and *B. cinerea* [37]. In other studies, isolates of *P. fluorescens* were able to inhibit *B. cinerea* in apples [38], or *Pseudomonas synxantha* showed the ability to control rots by *M. fructicola* and *M. fructigena* in peaches [39], together with many other examples for citrus [40], Chinese cherry [41], strawberry [42], and grapes [43].

In vitro tests showed that fungal growth was severely reduced around the bacterial colonies, suggesting that antifungal activity was performed by the metabolites produced by the bacterial isolates. Thus, to test this idea, we prepared an alcoholic extract from each of the three bacterial strains, and we tested the extracts for their performance in preventing the development of fungal pathogens, curing fungal diseases at early stage, and inducing resistance to fungal pathogen in fruits. Results show that extracts from our isolates were able to completely prevent decay by *P. digitatum* in tangerine and by *M. fructicola* in apricot, with a maximum disease incidence of ∼4%, while in tomato, the incidence of rots by *B. cinerea* reached a maximum of 25%. Similarly, the extracts from the same isolates were able to induce systemic resistance in apricots and tomatoes, and fully prevent the growth of the inoculated pathogens (*M. fructicola* and *B. cinerea*, respectively). In addition, the same isolates were able to cure early-stage infections (∼24 h from exposure) in tangerine fruits inoculated with *P. digitatum*, with a maximum disease incidence of ∼27%. This demonstrates that the *Pseudomonas* strains we isolated are able to produce metabolites that can act both directly and indirectly against fungal plant pathogens. The production of bioactive compounds from different strains of *Pseudomonas* has been previously reported [38,42,44,45,46]. Given the high bioactivity of the extracts from our isolates against fungal pathogens, we used untargeted metabolomics to attempt to identify the metabolites responsible of their efficacy. Indeed, we identified a pool of 71 metabolites common to all three bacterial isolates, but also a pool of unique metabolites produced by each of the three strains. Although we were only able to identify a few compounds, we found the presence of altenusin-like and phenazine-1-carboxylic acid-like molecules, both previously reported to have antifungal activity [47,48].

Our results pose interesting perspectives on using metabolites extracted from these bacteria as a tool for controlling post-harvest fungal pathogens. To follow on this idea, we set up three experiments where we simulated the use of our extracts in situations close to the actual needs of pre- and post-harvest operators. In the first trials, we tested the efficacy of the extracts against rots caused by *Colletotrichum* sp. on olives, and the results showed that they were able to reduce the incidence of the disease from 100% (control) to about 25%. Given the highest performance of the isolate B09 in most of the preliminary assays, we focused the other two field trials only on this isolate. In the second trial, we evaluated the efficacy of the extracts from isolate B09 in protecting sweet cherries from post-harvest decay mainly caused by *M. fructicola*. Results suggest a reduction of disease incidence and severity from 100% (control) down to 15–30%, according to the variety. As the last field trial, we tested the extract from isolate B09 directly in a vineyard, investigating its potential in controlling damages caused by powdery mildew in fruits and leaves of grapes. Likewise, in this case, extracts from our isolate proved to have high potential, and they effectively controlled powdery mildew on old leaves (those that directly received the sprayed formula), with results similar to the chemical control procedures normally performed by farmers. Only a few previous studies tested the efficacy of *Pseudomonas* isolates or their metabolites in controlling fungal plant pathogens in field conditions. For example, Wang et al. [49] showed that the pre-harvest application of *P. fluorescens* significantly reduced decay caused by *Penicillium* spp. Post-harvest fumigations with volatile organic compounds produced by *P. fluorescens* resulted in the control of disease caused by *Penicillium* spp. in oranges [40], and by *B. cinerea* in grapes [43]. This suggests that a detailed chemical characterization of the extracts used in this study might unveil the single or multiple molecules responsible for their bioactivity, opening new possibilities for the control of plant diseases. This also poses interesting perspectives on understanding the underlying molecular processes that regulate the production of these metabolites, and on identifying markers that can help in the selection of bacterial strains able to produce higher amounts, and thus likely to show higher performance as biocontrol agents.

In addition to our tests on *Pseudomonas* strains isolated from soil, for this paper, we often used the pomegranate peel extract (PGE) as a positive control. This product has been extensively tested by our group [23,24,27,50], and the results showed above further strengthen the possibility of using this extract as an alternative tool for plant disease control, perhaps in combination with other techniques such as the bacterial extracts used in this study.

Agriculture is at a turning point. Consumers are aware of the damages that chemical treatments cause to themselves and the environment, and this creates an increasingly higher amount of requests for produce that has been grown using sustainable approaches. Our study shows that by exploring the soil biodiversity, it is possible to select bacterial strains that produce metabolites with a high potential in the pre- and post-harvest control of fungal diseases. This type of screening can help us to find microbial metabolites that can finally become competitive with synthetic molecules and put an end to the use of chemical pesticides.

## Figures and Tables

**Figure 1 plants-11-00436-f001:**
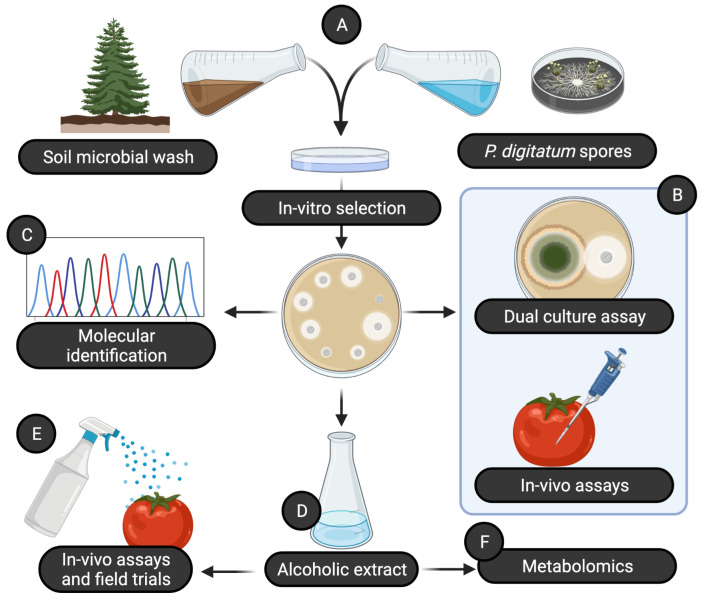
Workflow used in this study. (**A**) A microbial wash from environmental samples was mixed with a solution containing a high density of spores of *Penicillium digitatum*. (**B**) Isolates were screened for their efficacy using a wide set of in vitro and in vivo trials and (**C**) identified as *Pseudomonas* spp. by sequencing a portion of the 16S rRNA gene. From the most promising bacterial isolates, we produced a set of alcoholic extracts (**D**) that were then tested (**E**) for their efficacy in vivo and in setups simulating real-world applications, and (**F**) using an untargeted metabolomics approach, we attempted an annotation of the molecules likely to have antifungal activity. Created with BioRender.com.

**Figure 2 plants-11-00436-f002:**
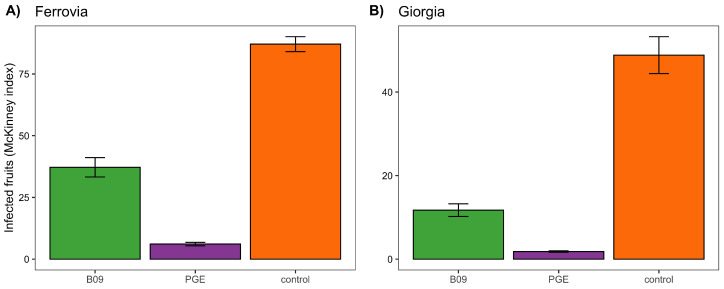
Efficacy of alcoholic extracts from bacterial isolates B01, B05, and B09 in reducing the incidence and severity (both accounted for using McKinney’s index) of fungal rot (mainly caused by *Monilinia fructicola*) on two varieties of sweet cherries: (**A**) Ferrovia and (**B**) Giorgia. The extract is compared to a plant extract with strong antifungal activity (pomegranate peel extract, PGE) and an untreated control group.

**Figure 3 plants-11-00436-f003:**
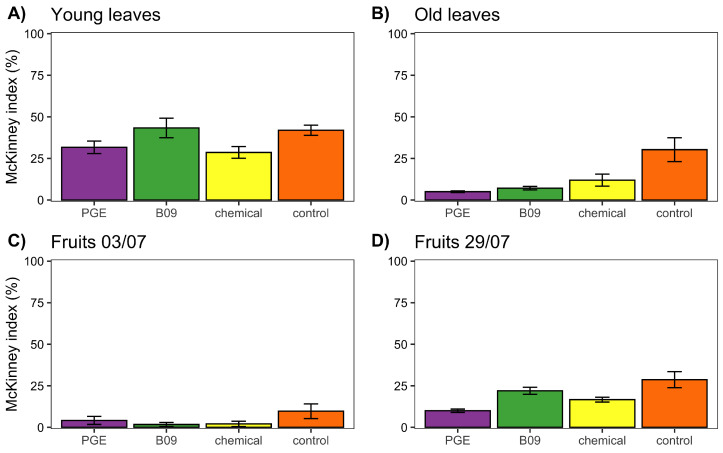
Efficacy of an alcoholic extract from the bacterial isolate B09 in reducing the incidence and severity (both accounted for using McKinney’s index) of the damage caused by powdery mildew on grapes in field conditions. The extract is compared to a plant extract with strong antifungal activity (pomegranate peel extract, PGE), the chemical treatments that are normally performed in the farm, and an untreated control group.

**Table 1 plants-11-00436-t001:** Summary of the host–pathogen combinations used for each trial. Tests were performed using alcoholic extracts from bacterial cultures, except the experiment marked with “*”, where we used live bacterial cultures.

Trial	Fruit	Variety	Pathogen
Preventive antifungal activity of live bacteria *	Apple	Golden delicious	*Penicillium expansum*
	Grape	Italia	*Botrytis cinerea*
	Olive	Ottobratica	*Colletotrichum acutatum*
	Tangerine	Avana	*Penicillium digitatum*
	Tomato	Datterino	*Alternaria alternata*
	Tomato	Datterino	*Botrytis cinerea*
Preventive antifungal activity of extracts	Apricot	Tsunami	*Monilinia fructicola*
	Tangerine	Avana	*Penicillium digitatum*
	Tomato	Datterino	*Botrytis cinerea*
Induction of resistance	Apricot	Tsunami	*Monilinia fructicola*
	Grape	Italia	*Botrytis cinerea*
	Tomato	Datterino	*Botrytis cinerea*
Curative effects	Tangerine	Avana	*Penicillium digitatum*
Post-harvest disease control on olives	Olive	Ottobratica	*Colletotrichum acutatum*
Post-harvest disease control on cherries	Sweet cherry	Ferrovia, Giorgia	*Monilinia fructicola*
Field trial	Grape	Gaglioppo	*Uncinula necator*

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
