# Peer review of "Extracts from Environmental Strains of Pseudomonas spp. Effectively Control Fungal Plant Diseases"

_plants, 2022, doi:10.3390/plants11030436_

Round 1

Reviewer 1 Report

According to the authors extracts from environmental strains of Pseudomonas spp. effectively control fungal plant diseases and from these isolates have remarkable biocontrol activity and may be employed effectively to control plant diseases in operational settings, demonstrating the potential of this strain.
The findings of this research demonstrate that the ambient microbiome is a key source of microorganisms that produce metabolites that have the potential to serve as an alternative to manmade chemical compounds.

This manuscript is well written and therefore it can be accepted for publication in the present form, just after a careful English language check.

Author Response

Dear Reviewer,

Thank you for you nice feedback. We have carefully revised the English style according to your suggestion.

Reviewer 2 Report

The author must follow the following comments:

The manuscript is interesting and fits within the scope of plants.

The relevant aspects of the topic are present.

I suggest some improvements in my review below an in the attached Pdf version

  • Title: the title needs to be changed as follow:
  • Extracts from environmental strains of Pseudomonas spp. effectively control fungal post-harvest diseases
  • The affiliation need to translate to English
  • In the Abstract, see line 18

Introduction:

  • Please see line 29
  • Please see line 38
  • Material and Methods, please see line 58
  • Please see line 67, What are these fungal pathogens?
  • Title of table 1 (please put the title in the front of the table)
  • please see line 123, (LB) what does it mean?
  • please see line 129
  • please see line 188
  • Please see line 243, What is the area of each plot and how many trees in each treatment?
  • please see line 390 What are the species that belong to this genus?
  • please see lines 429-432 remove this sentence from the discussion
  • please see line 445, remove the extra brackets

This discussion section is quite detailed and discusses well results of the study compared to the previous studies.

In Conclusion: The most relevant results are presented.

Best regards

Author Response

Dear Reviewer,

thanks for your comments. We carefully revised our manuscript following all your suggestions, except those not complying with the Journal's style. Also, given that our extracts were tested in field conditions, we prefer to keep the title as it is, rather than using the term "post-harvest". We included all the citations you suggested to add. We hope that you will now find the manuscript suitable for publication.

Reviewer 3 Report

The paper subject is interesting and can have large applications in biologic agriculture. In actual context, in which chemical pesticides contaminate soil, water, and crops, the market request for this type of  products  them can be huge. 
From my point of view, the bioproducts developed by authors can represent a real alternative to actual pesticides obtained from chemical synthesis.

Now, regarding the structure of the article:

1) The pieces of information are too much and not so well organized and presented. The readers are captivated by the subject but the quantity of information are huge.

Personal I think that the material presented here can be divided into   two articles, with the connection between them:

-part one (  the article one): which  will comprised:  in vitro screening regarding antifungal activities of some microbial isolates from soil  ( tests in Petri plates); preventive antifungal activity of alive bacteria; preventive antifungal activity of extracts; bacterial strains selection;   molecular analysis regarding microbial species selected; metabolomic studies regarding compounds found in bioproducts (extracts) responsible for antifungal effects.  

-part 2 (the article two): studies performed in vivo (on plants, fruits, and culture in the field)  which will comprise: Induction of resistance;  curative effects;  control of post-harvest rots on olives; control of post-harvest rots on sweet cherries; control of powdery mildew in a commercial grapevine orchard).

2)The authors tested their extracts in vitro ( antifungal activities are presented only by words without pictures or tables with results; indeed these were presented in supplementary materials, but the figure S1  is hard to read it. In my opinion, the figures and all materials from the supplementary source must be introduced in the  article(s).

3) Conclusions chapter missing

Recommendation to authors: divide this material into two parts, organize these as two articles, and resend these.

Author Response

Dear Reviewer,

thanks for your nice feedback. Although we agree with your that we presented a lot of information, the aim of this work is to present the entire workflow from selecting biocontrol agents to their application in field condition. Thus, we do not think it is appropriate to split it in two separate articles.

Also, the information we transferred to supplementary material are not really necessary to support our conclusions. Those results are included to support our selection of the best microbial strains, and transferring an extra figure and a huge table to the main manuscript would make it cluttered in our opinion. Readers can always refer to the supplementary material to have a look at these information if they feel they want to know more.

Conclusions are presented as the last paragraph of discussion, as per Plants instructions for authors that indicate "conclusions" as optional section.

We hope that you now will consider our revised manuscript suitable for publication.